# Tumor Cell-Associated IL-1α Affects Breast Cancer Progression and Metastasis in Mice through Manipulation of the Tumor Immune Microenvironment

**DOI:** 10.3390/ijms25073950

**Published:** 2024-04-02

**Authors:** Mathumathi Krishnamohan, Irena Kaplanov, Sapir Maudi-Boker, Muhammad Yousef, Noy Machluf-Katz, Idan Cohen, Moshe Elkabets, Jaison Titus, Marina Bersudsky, Ron N. Apte, Elena Voronov, Alex Braiman

**Affiliations:** 1The Shraga Segal Department of Microbiology, Immunology and Genetics, Faculty of Health Sciences, Ben-Gurion University of the Negev, Beer Sheva 84105, Israel; mathumathi00@gmail.com (M.K.); moshee@bgu.ac.il (M.E.); jaisontitus93@gmail.com (J.T.); marinab@bgu.ac.il (M.B.);; 2Cancer Center, Emek Medical Center, Afula 18101, Israel; idan5161@gmail.com

**Keywords:** interleukin-1-alpha, inflammation, intracellular IL-1α, metastasis, MDSC

## Abstract

IL-1α is a dual function cytokine that affects inflammatory and immune responses and plays a pivotal role in cancer. The effects of intracellular IL-1α on the development of triple negative breast cancer (TNBC) in mice were assessed using the CRISPR/Cas9 system to suppress IL-1α expression in 4T1 breast cancer cells. Knockout of IL-1α in 4T1 cells modified expression of multiple genes, including downregulation of cytokines and chemokines involved in the recruitment of tumor-associated pro-inflammatory cells. Orthotopical injection of IL-1α knockout (KO) 4T1 cells into BALB/c mice led to a significant decrease in local tumor growth and lung metastases, compared to injection of wild-type 4T1 (4T1/WT) cells. Neutrophils and myeloid-derived suppressor cells were abundant in tumors developing after injection of 4T1/WT cells, whereas more antigen-presenting cells were observed in the tumor microenvironment after injection of IL-1α KO 4T1 cells. This switch correlated with increased infiltration of CD3^+^CD8^+^ and NKp46^+^cells. Engraftment of IL-1α knockout 4T1 cells into immunodeficient NOD.SCID mice resulted in more rapid tumor growth, with increased lung metastasis in comparison to engraftment of 4T1/WT cells. Our results suggest that tumor-associated IL-1α is involved in TNBC progression in mice by modulating the interplay between immunosuppressive pro-inflammatory cells vs. antigen-presenting and cytotoxic cells.

## 1. Introduction

Triple-negative breast cancer (TNBC) is the most malignant form of breast cancer, with a high metastatic potential. TNBC is the leading cause of female cancer-related death [1,2,3,4]. Inflammation has been shown to be a main player in the development of many tumors, including breast cancer [5,6,7]. Pro-inflammatory cytokines, such as interleukins, promote inflammation in the tumor microenvironment (TME) and exacerbate cancer cell invasion [8,9,10,11,12]. IL-1α is an upstream cytokine that is constitutively expressed in a variety of cells and plays a pivotal role in inflammation and cancer [8,13,14]. IL-1α is translated into a 31 kDa precursor form (pro-IL-1α) that can be processed by calpain into the mature 17 kD form and the 16 kD N-terminal cleavage product—IL-1α propiece. However, in most cases, IL-1α remains unprocessed and is retained intracellularly or as a membrane-associated protein [15]. Both pro-IL-1α and the propiece contain an active nuclear localization sequence (NLS) in their N-terminal region, which drives its predominantly nuclear localization [8]. It has been shown that nuclear IL-1α can regulate cell proliferation, with conflicting reports of its pro- or anti-proliferative effect, depending on the cell type tested [13]. Other studies of IL-1α suggested several nuclear functions, including interaction with the spliceosome components [16], and regulation of transcription [17,18,19]. It has been shown that during apoptosis, intracellular pro-IL-1α binds tightly to chromatin; while during necrosis, the precursor dissociates from the chromatin and is released from the cell into the microenvironment, where it serves as an “alarmin” that initiates inflammation [8,20,21,22]. Released IL-1α activates IL-1 receptor (IL-1R1) and induces expression of other pro-inflammatory cytokines and chemokine secretion, and recruitment of neutrophils that intensifies local inflammation [23]. For example, in the model of dextran sulfate sodium-induced colitis, we confirmed that IL-1α plays an essential role in supporting colon inflammation [24]. Moreover, IL-1α promotes the development of diverse chronic inflammatory diseases and is tightly involved in tumor development [8,14,25].

It was found that higher levels of IL-1α correlates with increased metastasis and a significantly lower five-year survival rate in patients with head and neck squamous cell carcinoma [26]. In melanoma patients, IL-1α was expressed in 73% of inspected nevi, most primary tumors (98%), and in more than a half of metastatic lesions (55%) [27].

Due to its unique properties and possibility to be active as an intracellular, membrane-associated, and secreted molecule, IL-1α can produce both pro- and anti-tumor effects [8]. For example, in the MMT-Her2 model of breast carcinoma, IL-1α deficiency delayed tumor development, probably by reduction of inflammatory microenvironment, and by reduction of cancer stem cells (CSCs) in primary tumors [28]. A strong correlation between the expression of IL-1α and the CSC-positive phenotype has been observed in clinical specimens from breast cancer tumors [29]. IL-1α can increase matrix metalloproteinase expression and induce angiogenesis in growing tumors, albeit less efficiently than IL-1β [30,31,32]. Yet, other studies have shown that overexpression of IL-1α may have anti-tumorigenic effects [14,33,34,35]. This body of evidence indicates the necessity for a more careful evaluation of the biology of IL-1α in TNBC development in order to design novel therapeutic approaches for this severe form of breast cancer that is resistant to conventional treatment.

It has been shown in numerous studies on human breast cancers and in murine models that immune cells are the main players in the building, shaping, and reprogramming of the tumor microenvironment [9,10,14,36,37,38,39,40,41]. Myeloid cells, especially macrophages, are a major cellular component of the TME [42,43,44,45,46,47,48,49,50]. Cells with both pro-and anti-tumorigenic functions can co-exist in the same tumor, and the balance between these cells determines the fate of the malignant process [42,51]. Recently we have found that IL-1β, which is another IL-1 agonistic molecule, can be involved in recruitment and differentiation of myeloid cells in the TME of 4T1 injected mice [52]. While in control BALB/c mice heavy infiltration of pro-inflammatory macrophages was observed, the TME in IL-1β-deficient mice was characterized by low levels of macrophages and a relatively high percentage of CD11b^+^ dendritic cells (DCs). It should be noted that unlike IL-1α, which is active both intra- and extracellularly and is expressed by both malignant and the TME cells, IL-1β is active only in its secreted form and is normally produced only by the immune cells [12]. 

In this study, we show that both tumor- and the TME-derived IL-1α is involved in tumor development in mice. IL-1α deficiency in TNBC leads to a significant decrease in tumor growth in immunocompetent mice, probably due to a reduction in inflammation and modulation of immune cells in the TME. IL-1α knock-out in breast cancer cells led to diminished tumor infiltration by myeloid-derived suppressor cells (MDSC), and to increased recruitment of antigen-presenting and cytotoxic cells into the tumor site. However, this knockout failed to suppress tumor development in immunodeficient mice. These findings indicate that tumor cell-associated IL-1α is involved in regulation of the tumor immune microenvironment, and that IL-1α knocked-out cells become more immunogenic. Additionally, our results indicate that elimination of the microenvironment-derived IL-1α also inhibits the growth of breast cancer grafts in mice. Thus, both tumor- and the TME-derived IL-1α contribute to the progression of TNBC in mice.

## 2. Results

### 2.1. Tumor-Associated Intracellular IL-1α Is Involved in TNBC Growth in Mice

The existing body of evidence indicates that IL-1α production in cancer cells can play a major role in tumor progression [8]. In this study, we used the mouse mammary carcinoma cell line 4T1, a highly tumorigenic, poorly immunogenic, invasive, and spontaneously metastatic line, which is one of the most widely used murine models of TNBC [53,54]. IL-1α is constitutively expressed in resting 4T1 cells and is primarily localized in the nucleus (Appendix A). Moreover, pretreatment of the cells with either LPS or TNFα, well-known pro-inflammatory inducers of IL-1, did not affect the level of IL-1α expression in 4T1 cells (Appendix A). We could not detect IL-1α expression either on the plasma membrane of unpermeabilized 4T1 cells using immunostaining or in 4T1-conditioned medium using ELISA (Appendix A). 

To assess if IL-1α deficiency in tumor cells affects tumor progression in mice, we created IL-1α knock-out 4T1 cells (4T1 IL-1α KO) using the CRISPR/Cas9 approach. The absence of IL-1α expression in the generated KO clones was verified by imaging, flow cytometry analyses, and qPCR (Appendix A). The clones, which showed negligible levels of IL-1α expression, were further subjected to DNA sequencing of the genomic region targeted by the CRISPR/Cas9 system to confirm that mutations were created in the relevant regions of the *IL-1α* gene (Appendix A). Three distinct KO clones, showing negligible IL-1α expression levels and abnormal DNA sequences in the gene area corresponding to one of the sgRNAs, were selected for further experiments. 

To assess the functional significance of IL-1α expression in TNBC cells for tumor growth, Cas9-expressing 4T1 (4T1/WT) and IL-1α KO cells were injected orthotopically into the mammary fat pad of BALB/C mice. In mice injected with 4T1/WT cells, tumor growth was apparent by day 15, developing into large tumors by day 30; whereas mice injected with either IL-1α KO clone showed much slower tumor progression (Figure 1A–C). Moreover, only about 60% of mice injected with IL-1α KO cells developed small tumors within the timeframe of the experiment (Figure 1D). 

Tumor-bearing mice developed spontaneous lung metastases by day 32–35. Significantly higher numbers of both micro- and macro-metastases were observed in lungs obtained from 4T1/WT tumor-bearing mice, in comparison to lungs from mice injected with 4T1 IL-1α KO cells, some of which remained metastasis-free until day 39 (Figure 1E,F).

### 2.2. Deficiency of IL-1α Leads to Changes in 4T1 Phenotype In Vitro

Given the profound effect of knocking out the *IL-1α* gene on the growth of 4T1 cells in vivo, we examined if *IL-1α* deficiency had any effect on tumorigenicity-related properties of these cells when grown in culture. One of the hallmarks of cancer is the ability of malignant cells to survive under metabolic stress and serum independence [55]. We found that the proliferation rate of 4T1 cells was not affected by knocking out IL-1α under normal growth conditions. However, 4T1 IL-1α KO cells were significantly more resistant to stressful conditions, such as serum (Figure 2A–D) or glucose (Figure 2E) starvation, than 4T1/WT cells. Additionally, IL-1α KO cells were less adherent and more migratory, yet showed less colony formation potential in comparison to 4T1/WT cells, probably due to the reduced adherence (Figure 2F–H).

Taken together, these results suggest that IL-1α KO cells exhibit a more aggressive phenotype in vitro, which ostensibly should have led to a more rapid tumor growth and extensive metastases in vivo. This prediction stands in striking contrast to the results shown in Figure 1. To address this apparent discrepancy, a deeper evaluation of 4T1/WT vs. 4T1 IL-1α KO cells had to be conducted, as shown and discussed below.

### 2.3. Deletion of IL-1α Affects Gene Expression in 4T1 Cells

It has previously been shown that intranuclear IL-1α can bind chromatin and may participate in the regulation of gene expression [17,19,20,56]. To assess the transcription regulation of IL-1α in 4T1 cells, we performed total mRNA sequencing (RNAseq) in 4T1/WT and 4T1 IL-1α KO cells. Three distinct clones of 4T1 IL-1α KO and Cas9-expressing 4T1/WT cells were sequenced with three technical replicates for each cell sample. The bioinformatics analysis of the results identified 13,691 distinct mRNA transcripts in total. Of these, 820 were either significantly upregulated or downregulated (fold > 1.5, adjusted *p*-value < 0.05) in 4T1 IL-1α KO versus 4T1/WT cells (Figure 3A). KEGG mapping of significant genes revealed enrichment of pathways associated with cancer, cell proliferation and survival, cytokine network function, cell motility and adhesion (Figure 3B). These findings correlated with the phenotypical alterations in 4T1 IL-1α KO cells shown above (Figure 2). Selected differentially expressed genes linked to the enriched pathway are shown in (Figure 3C,D). The full list is shown in Appendix A. To confirm the RNAseq results we performed qPCR analysis of selected cytokines and chemokines (Figure 3E–J, Appendix A), which have been shown to affect tumor development and inflammation [8,9,10,11,12,37,38,57,58]. Indeed, we found that knockout of IL-1α in 4T1 cells affected the expression of several cytokines, chemokines, and other factors associated with tumor growth and progression. For example, expression of the typical pro-inflammatory factors, such as IL-6 and GM-CSF, was downregulated in 4T1 IL-1α KO vs. 4T1/WT cells (Figure 3C,E,F). On the other hand, expression of the typically anti-inflammatory TGFβ was elevated in 4T1 IL-1α KO cells (Figure 3G). These changes suggest a reduction in tumorigenic potential of 4T1 cells following knockout of IL-1α. Interestingly, 4T1 IL-1α KO cells also exhibited elevated levels of TSLP (Figure 3D and Appendix A), PGDFα and VEGFα expression (Figure 3H,I). These factors are normally associated with tumor growth promotion and increased angiogenesis. Thus, the effect of intracellular IL-1α on the malignant potential of TNBC cells may not be straightforward. In agreement with the RNAseq results, we found that cells deficient in IL-1α express higher levels of PCNA in comparison to 4T1/WT cells (Figure 3D and Appendix A). Expression of the pro-survival factors Bcl-2 and Bcl2l1 was also increased in 4T1 IL-1α KO cells (Figure 3D and Appendix A). These may account for the increased proliferative capacity in 4T1 IL-1α KO cells under suboptimal conditions (Figure 2). Our results also show reduced expression of chemokines in 4T1 IL-1α KO cells (Figure 3J and Appendix A), which could potentially lead to a diminished recruitment of immune cells to the tumor. Finally, our findings indicate that intracellular IL-1α can also be involved in regulation of epithelial–mesenchymal transformation (EMT), since the expression levels of Zeb2, e-cadherin, β-catenin, vimentin, and Twist1 are altered in 4T1 IL-1α KO cells in a pattern that favors EMT (Figure 3D and Appendix A) [59]. Partial EMT can account for the more motile and less adhesive phenotype of IL-1α KO cells (Figure 2), as well as pointing toward a more aggressive and metastatic tumor. 

### 2.4. Expression of IL-1α in 4T1 Cells Affects the TME

To evaluate the nature of immune cells recruited into tumors following inoculation of mice with either WT/4T1 cells or 4T1 IL-1α KO cells, the tumor cellular composition was analyzed using flow cytometry (FC) at different time points post tumor cell inoculation. 4T1 IL-1α KO-derived tumors exhibited lower levels of the total myeloid (CD11b^+^) and MDSC populations (Figure 4A,B), but much higher levels of activated antigen-presenting cells (APC) (Figure 4C). The differences were more pronounced at later time points. This trend was confirmed by IHC staining, which showed higher levels of neutrophil infiltration (Figure 4D), but lower numbers of mature macrophages (F4/80^+^) in 4T1/WT-derived tumors (Figure 4E).

In addition, we observed dramatic differences in cytotoxic cell populations in the TME of the two groups. Flow cytometry analysis revealed an abundance of CD3^+^CD8^+^ (CTLs) and NKp46^+^ (NK) cells in 4T1 IL-1α KO-derived tumors, when compared to WT/4T1 tumors (Figure 4F,G). In fact, CTL infiltration foci could be detected in 4T1 IL-1α KO-derived tumors already on day 4 after tumor cell injection (Figure 4H). These results correlated with flow cytometry analyses, which revealed a significant increase in CTLs in the TME from mice injected with 4T1 IL-1α KO cells at different time intervals (Figure 4F). 

All mice injected with 4T1/WT cells developed profound splenomegaly, while in mice injected with 4T1 IL-1α KO cells the changes in spleen size were mild (Appendix A). Starting on day 25 post-inoculation, the numbers of B and T cells began to drop in spleens of mice injected with 4T1/WT cells (Appendix A). This was accompanied by a strong increase in the number of myeloid CD11b^+^ cells and MDSC in the spleens of these mice (Appendix A). In mice injected with 4T1 IL-1α KO cells, T and B cell populations were more preserved and significantly fewer numbers of pro-inflammatory myeloid cells were observed (Appendix A). The changes in cellular composition were highly correlated with splenomegaly development. 

Given the substantial differences in the cellular composition in 4T1/WT vs. 4T1 IL-1α KO tumors, we assessed the expression of key immunomodulatory molecules and growth factors in tumors obtained from both groups of mice. As summarized in Figure 4I (qPCR) and Figure 4J–L (ELISA), tumor tissues showed a significant decrease in most pro-inflammatory cytokines and pro-angiogenic factors in the TME after injection of IL-1α KO tumor cells. However, Granzyme A and IFNγ were markedly increased, corresponding to the increased tumor infiltration by CTLs and NK cells. While IL-1α KO tumors contained lower levels of chemokines, such as CCL2 and CXCL1/2/3, known to attract MDSCs into the tumor [10,57,60], they showed higher expression of CCL5, CXCL9/10, which could attract T lymphocytes and NK cells [61] (Figure 4I). 

Overall, these results demonstrate that tumor-associated IL-1α leads to more intensive recruitment of immature suppressive myeloid cells, which secrete pro-inflammatory cytokines and support tumor progression. On the other hand, 4T1 IL-1α KO-derived tumors were characterized by less inflammation and an abundance of antigen-presenting and cytotoxic cells, which could be responsible for the inhibition of tumor growth. 

### 2.5. Inhibition of Tumor Development after Injection of 4T1 IL-1α KO Cells Is Critically Dependent on the Host Immune Response

To confirm that tumor growth inhibition in mice injected with IL-1α deficient 4T1 cells was dependent on the host immune response, we injected either 4T1/WT or 4T1 IL-1α KO cells into immunocompromised NOD.SCID mice and compared tumor growth with those from immunocompetent BALB/c mice. In NOD.SCID mice, both types of cells, with and without IL-1α, induced rapid tumor growth (Figure 5). The tumors became detectable on day 12, and by day 25 the tumors in the immunocompromised mice were significantly larger than those in the immunocompetent mice (Figure 5A). Remarkably, 4T1 IL-1α KO-derived tumors were even more aggressive than 4T1/WT-derived tumors in the immunocompromised mice. The dynamics of tumor growth correlated with tumor weights (Figure 5B). Immunocompromised mice exhibited less splenomegaly in comparison to immunocompetent mice. Nevertheless, in NOD.SCID mice injected with IL-1α KO cells, the spleens were enlarged, compared to those from 4T1/WT tumor-bearing mice (Figure 5C). Multiple metastatic nodules were observed in lungs of immunocompromised mice on day 25, while virtually no metastases were detected in lungs of BALB/c mice at the same time point. The metastatic count was significantly higher in the lungs of NOD.SCID mice that were injected with 4T1 IL-1α KO (Figure 5D–F). Similar results were obtained using immunodeficient NOD.SCID/IL2rγ^null^ (NSG) mice (Appendix A), where the described effects were even more pronounced.

In contrast to immunocompetent mice, NOD.SCID mice exhibited significantly higher myeloid infiltration in tumors developed from 4T1 IL-1α KO vs. 4T1/WT cells (Figure 6A). On the other hand, immunocompetent and immunodeficient mice showed similar patterns of NK tumor infiltration (Figure 6B). Appropriately to the NOD.SCID phenotype, no CD3^+^ or MHCII^+^ cells were detected in tumors derived from these mice. To further confirm the role of cellular immunity in the suppression of 4T1 IL-1α KO-induced tumor growth, we performed a Winn assay. Either BALB/c or NSG mice were inoculated with 4T1 IL-1α KO cells. On day 25 after cell injection, spleens were excised and the derived splenocytes were injected, together with 4T1/WT cells, into BALB/c mice. We observed some retardation of tumor growth only in mice that received adoptive transfer of spleen cells from 4T1 IL-1α KO tumor-bearing BALB/c mice, but not from NSG mice (Figure 6C).

Altogether, these results confirmed the critical role of the host immune response in the inhibition of tumor growth in mice inoculated with 4T1 IL-1α KO cells. Moreover, once the immune suppression was excluded, 4T1 IL-1α KO cells could fully realize their enhanced malignant potential, compared to 4T1/WT cells.

### 2.6. Both the Microenvironment-Derived and Tumor-Derived IL-1α Affect 4T1-Induced Tumor Development in Mice

The existing body of evidence suggests that both tumor- and TME-derived IL-1α can affect tumor development [8,25,30]. To assess the possible impact of host-derived IL-1α- and IL-1R1-dependent signaling on the phenomena presented here, control BALB/c, IL-1α KO mice and IL-1Ra KO mice were inoculated with either 4T1/WT or 4T1 IL-1α KO cells, as described above.

In IL-1α KO mice, tumors derived from each cell line developed significantly slower than tumors derived from the same cell line in WT BALB/c mice. In fact, cells deficient in IL-1α that were injected into IL-1α KO mice did not grow at all, while tumor development induced by 4T1/WT cells was strongly suppressed (Figure 7A,B). Remarkably, tumor growth in IL-1Ra KO mice injected with IL-1α KO cells was strongly inhibited, in comparison to injection of 4T1/WT cells, despite the pronounced pro-inflammatory TME phenotype of these mice (Figure 7C,D). These findings further indicate that both tumor cell-associated and the TME-derived IL-1α contribute significantly to tumor development.

To separate the effects of extracellular vs. intracellular IL-1α on tumor development, we injected 4T1/WT-inoculated BALB/c mice with neutralizing anti-IL-1α antibodies, thereby blocking the effects of extracellular IL-1α. However, neutralization of extracellular IL-1α could not recapitulate the strong tumor suppression induced by IL-1α KO. Anti-IL-1α treatment resulted in a moderate attenuation of tumor growth (Figure 7E), with no significant reduction in mortality (Figure 7F). 

## 3. Discussion

The important role of pro-inflammatory cytokines, including IL-1α, in invasiveness of malignant cells in cancer patients and in different experimental tumors in mice has previously been shown [5,6,7,8,36,62]. IL-1α can regulate cell proliferation, with conflicting reports of its pro- or anti-proliferative effect, depending on the cell type tested [13]. Overexpression of IL-1α can trigger malignant transformation or promote cell survival; yet, it has also been shown to induce apoptosis in malignant cells [13,16,63,64]. 

In most tumors, IL-1α expression and secretion correlated tightly with progressive growth and metastasis development [26,65,66,67,68]. However, in other instances, IL-1α expression correlated with a good prognosis [68,69,70]. In light of these conflicting results, the immune regulatory role of IL-1α in tumor development must be studied in more detail. 

The main goal of this study was to evaluate the role of IL-1α expression in tumor cells in the progression of TNBC in mice. We therefore designed IL-1α knockout 4T1 breast cancer cells using the CRISPR/Cas9 approach (Appendix A). 

Our results demonstrate that *IL-1α* gene knockout induces substantial changes in the transcriptome of 4T1 breast cancer cells (Figure 3, Appendix A). We observed enrichment of pathways associated with cancer, cell proliferation and survival, cytokine network function, cell motility and adhesion after knockout of IL-1α from tumor cells. It is notable that these pathways are also the main targets of extra-cellular IL-1 molecules acting via IL-1R1 [71]. Our results indicate that depletion of intracellular IL-1α in 4T1 cells leads to a reduction in expression and secretion of various pro-inflammatory molecules (Figure 3, Appendix A). It has been reported that secreted IL-1α is a powerful driver of Type 3 immunity, sustaining polarization of innate and adaptive lymphoid cells in the ILC3 and Th17 direction [72]. Here, we show that knocking out IL-1α in 4T1 cells results in the downregulation of IL-17RE and IL-23 (Figure 3). On the other hand, increased expression of TSLP and IL-33 in IL-1α KO cells may promote Th2 polarization. These changes, together with increased expression of TGF-β and downregulation of GM-CSF and IL-6, suggest that the lack of IL-1α production in 4T1 cells shifts the cytokine network balance toward a more anti-inflammatory state that subsequently leads to a less tumorigenic environment in vivo [8]. It should be noted that although Th2 polarization is generally associated with inhibition of anti-tumor immune responses, it has also been demonstrated that tumoral expression of IL-33 inhibits tumor growth and promotes activity of CD8^+^ T and NK cells in the TME [73,74].

In addition to changes in the cytokine network pathway, IL-1α deficiency in 4T1 cells resulted in the downregulation of multiple molecules, such as integrins, laminins, EpCAM, E-cadherin and tight junction molecules, which are responsible for cell adhesion, cell-extra cellular matrix interactions, and cell motility (Figure 3, Appendix A). Functional experiments demonstrated that IL-1α KO cells acquired a less adhesive and more mobile phenotype (Figure 2). Together with the upregulation of vimentin, β-catenin, Zeb2 and Twist1 (Appendix A), these changes are consistent with EMT alterations [59]. Additionally, elevated expression of PDGF, VEGF and TGFβ suggests a higher proliferative and angiogenic potential in IL-1α KO cells. Reduced serum and glucose dependency (Figure 2), as well as upregulation of BCL-2, Bcl2l1 and PCNA, support this conclusion (Figure 3, Appendix A). These findings point to a more invasive phenotype of 4T1 IL-1α KO cells than that of 4T1/WT cells. 

Thus, the effects produced by IL-1α deficiency in 4T1 breast cancer cells in culture are a complex combination of both potentially pro- and anti-tumorigenic phenomena. Observing growth of WT and KO cells in vivo was necessary to determine the net results of these alterations. Surprisingly, our data showed that the tumorigenicity and metastatic properties of 4T1 IL-1α KO cells in BALB/c mice was dramatically reduced, in comparison to WT/4T1 cells (Figure 1). To resolve this apparent contradiction, a more extensive investigation of 4T1 IL-1α KO tumor properties had to be undertaken.

Early immune responses in the tumor microenvironment are key to determining the disease outcome. Thus, we examined cellular composition of tumors from mice injected with WT/4T1 versus 4T1 IL-1α KO cells at different time intervals. There were no evident differences in tumor growth between the two groups up to approximately day 14; nevertheless, even at these early time intervals, increased recruitment of immune cells into 4T1 IL-1α KO tumors was observed (Figure 4). Significant differences in the total number of myeloid (CD11b^+^) cells, MDSCs and mature macrophages (F4/80^+^ cells) were evident at later time points. Additionally, we observed higher levels of cytotoxic T (CD3^+^CD8^+^) cells, NK (NKp46^+^) cells and activated APCs, essential for T cell activation in the TME of 4T1 IL-1α KO tumors. These changes correlated with the increased expression of Granzyme A and INFγ (Figure 4).

Tumor-associated macrophages (TAMs) are among the most important cells in tumor progression. Breast cancer is characterized by the presence of a large population of TAMs [43,44,52,75], which are a key component of the tumor stroma and are essential for angiogenesis and matrix remodeling that supports progressively growing tumors. Tumor-associated neutrophils have also been shown to play an important function in tumor-bearing mice and cancer patients [76,77,78,79]. It has also been shown that IL-1R signaling is involved in MDSCs expansion, migration and function [46,48,80], and that IL-1α secreted from a tumor can suppress T cell function via promoting MDSCs in the microenvironment of HCC [47,68]. 

Since the cellular composition of the TME had been clearly affected by IL-1α, it made sense to assess chemokine levels in tumor tissues. Our results showed that 4T1/WT tumors contained higher levels of CXCR2 ligands, such as CXCL1/2/3, known to attract MDSCs into the tumor, thereby enhancing cancer cell survival [10,57,60] (Figure 4). However, the TME of IL-1α KO tumors showed higher expression of CXCR3 ligands, such as CXCL9/10, which can attract anti-tumor CXCR3^+^ CTL, and Th1 lymphocytes, and NK cells [61].

The progression of 4T1/WT tumors in mice strongly correlated with splenomegaly and profound changes in the white pulp cellular composition of spleens (Appendix A), due to the involvement of IL-1 molecules in extramedullary hematopoiesis [81]. However, in mice injected with 4T1 IL-1α KO cells, splenomegaly was not pronounced, and the cellular infiltrate was well preserved (Appendix A).

The results described above suggested that despite the higher malignant potential of 4T1 IL-1α KO cells observed in vitro, the immune microenvironment within IL-1α KO tumors was less favorable for tumor progression. Thus, an enhanced immune response against 4T1 IL-1α KO cells could explain the suppressed growth of these cells in vivo. To test this hypothesis, we introduced 4T1/WT and 4T1 IL-1α KO cells into immunodeficient NOD.SCID and NSG mice, strains depleted of both T and B cells, and lacking MHC-II expression. The pattern of tumor development observed in immunodeficient mice was the opposite of that seen in WT mice (Figure 5 and Figure 6, Appendix A), i.e., injection of IL-1α KO cells into immunodeficient mice resulted in a more rapid tumor growth and an extensive metastatic spread, compared to immunodeficient mice injected with 4T1/WT cells. These results support the conclusion that inhibition of tumor development in mice injected with 4T1 IL-1α KO cells is due to the augmented host immune response to these cells. 

Elkabets et al. showed that IL-1α KO fibrosarcoma cell lines are rejected in immunocompetent, but not in immunodeficient, mice [82]. This is in accord with our findings, showing uncontrolled growth of IL-1α KO breast cancer cells in immunocompromised mice (Figure 5, Appendix A) but limited growth in WT mice. In the absence of anti-tumor immune cells, TNBC cells could grow unimpeded, with IL-1α KO cells fully realizing their increased aggressive potential, which was observed in the in vitro experiments (Figure 2), and exceeding the rate of tumor progression achieved by 4T1/WT cells.

Contrary to our findings showing apparent inhibition of anti-tumor immune response by IL-1α, it has also been demonstrated that the membrane-associated form of IL-1α can be efficient in activating innate anti-tumor immune responses, by acting as a focused adjuvant [69]. Additionally, it has been shown that IL-1 receptor signaling by its extracellular ligands can also boost anti-tumor immunity, via recruitment and activation of antigen presenting cells to lymph nodes, where they can induce the activation, expansion and development of immunological memory in CD4^+^ and CD8^+^ T cells specific to tumor-associated antigens [25]. These results indicate that IL-1α of tumor origin can play a complex role in TNBC development, affecting both the malignant potential of the tumor cells and modulating the immune response to the tumor toward different outcomes. The end result of these effects is ultimately dependent on the interplay between the inflammatory properties of the cells, their invasiveness, location of IL-1α (intra or extracellular), and the host immune response.

To assess if the microenvironment associated IL-1α could also play a role in TNBC development, we engrafted 4T1 cells into IL-1α KO and IL-1Ra KO mice. The overall pattern of tumor development induced by 4T1 cells in these transgenic mice conformed with our previous results showing that IL-1 agonistic molecules in the TME play an important role in the development of tumors and metastases [23,30,52,69]. No tumor growth was observed following 4T1 IL-1α KO cell injection in IL-1α KO mice, while tumor development induced by 4T1/WT cells was significantly attenuated (Figure 7A,B). Moreover, IL-1α deficiency in tumor cells reduced tumor growth even in the highly pro-inflammatory microenvironment of IL-1Ra KO mice [83] (Figure 7C,D). Interestingly, the strongly augmented IL-1 receptor signaling observed in IL-1Ra KO mice did not significantly affect 4T1/WT tumor development in these animals (Figure 7C,D), suggesting the dominant effect of intracellular IL-1α, independently of IL-1 receptor activation.

Although IL-1α is mainly retained intracellularly, a naturally occurring human IL-1α neutralizing antibody MABp1 (XBiotech Inc., Austin, TX, USA) and an antibody targeting IL-1α (Bermekimab) have shown promising results with several types of cancers [14,84]. However, neutralization of extracellular IL-1α could not recapitulate the strong tumor suppression induced by IL-1α KO. Anti-IL-1α treatment resulted in a moderate attenuation of tumor growth (Figure 7E), with no significant reduction in mortality (Figure 7F). These results further emphasize the critical role of intracellular IL-1α in regulation of tumor growth and in the host immune response to it.

In conclusion, tumors actively influence the immune response through the production of factors that attract immune cells and subsequently alter their ability to recognize and effectively combat the tumor (Figure 8). Here, we demonstrate, for the first time, that intracellular IL-1α in breast cancer cells can alter the TME to ensure their survival via expression of pro-inflammatory molecules and recruitment of myeloid immunosuppressive cells. 4T1 IL-1α KO cells in vitro exhibit a more aggressive phenotype, probably due to higher survivability under challenging metabolic conditions, which usually occur in growing tumors, and partial EMT. Yet, 4T1 IL-1α KO cells evoke a much stronger anti-tumor immune response in an immune-competent host, which compensates for the increase in cell malignancy and suppresses tumor growth. Diminished recruitment of pro-tumorigenic myeloid cells, such as TAMs, neutrophils and MDSCs, together with enhanced recruitment of activated myeloid APC, CD8^+^ T and NK cells to the TME results in reduced TNBC development in mice (Figure 8). In addition, our results show that both tumor- and the TME-derived IL-1α contribute to the progression of TNBC in mice. Targeting intracellular IL-1α in cancer and/or host cells elicits the most pronounced effect on tumor growth. Suppression of IL-1α expression in tumor cells and the TME, e.g., using nanoparticle delivery systems [85], could be a promising strategy for battling breast cancer. 

## 4. Methods

### 4.1. Cell Lines

The mammary carcinoma cell line 4T1 was received and maintained, as described [52]. HEK 293T (RRID:CVCL_0063, CRL-3216, ATCC) were grown in complete DMEM (GIBCO), according to standard protocols.

### 4.2. Plasmids

pCW57.1 was a gift from David Root (Addgene plasmid #41393; RRID: Addgene_41393) and pLKO5.sgRNA.EFS.GFP was a gift from Benjamin Ebert (Addgene plasmid #57822; RRID:Addgene_57822). 

pHDM-Hgpm2 (Addgene plasmid #164441; RRID:Addgene_164441), 

pHDM-Tat1b (Addgene plasmid #164442; RRID:Addgene_164442), 

pHDM-VSV-G (Addgene plasmid #164440; RRID:Addgene_164440), 

pRC-CMV-Rev1b (Addgene plasmid #164443; RRID:Addgene_164443) were gifts from Alejandro Balazs.

pLKO5.sgRNA.EFS.GFP was digested using BsmBI and a pair of annealed oligos were cloned into the vector, replacing the sgRNA scaffold. The oligos were designed based on the target site sequence (20 bp) and required a 3 bp NGG PAM sequence at the 3’ end. Five alternative sgRNA pairs targeting different gene regions were created to ensure efficient gene knockout (Appendix A).

### 4.3. Cell Transfection/Infection

To transfect cells with the Cas9 plasmid and IL-1α construct plasmids, the lentiviral expression system was used. For this purpose, 4 × 10^6^ packaging cells (HEK293T) were seeded onto 100 mm petri dishes and incubated for 24 h at 37 °C, 5% CO_2_ in 10 mL DMEM.

The DNA mixture contained 15 µg Cas9 plasmid, 5 µg Gag-Pol, 3.5 µg VSV-G, 3 µg Rev, 3.5 µg Tat in 500 µL serum-free DMEM. The DNA solution was mixed with 500 mL serum-free DMEM containing 90 µg linear 25kD polyethylenimine (Polysciences, cat# 23966-2). The mixture was incubated for 15–20 min at room temperature, then carefully transferred to the packaging cells. The cells were incubated for 24 h. Afterwards, the medium was replaced with 10 mL of complete DMEM. At 48–72 h post transfection, the virus containing supernatant was collected, centrifuged at 10,000× *g*, and added to the target cells (4T1). The medium was replaced after 24 h and 5 mg/mL Blasticidin was added 48 h postinfection. Protein expression was assessed using immunostaining and flow cytometry.

The sgRNA bearing plasmids were transiently expressed in Cas9^+^ cells using TransIT^®^-2020 Transfection Reagent (Mirus), according to the manufacturer’s instructions. The GFP^+^ cells were sorted and collected into 96well plates, at 1 cell per well, to create potential KO clones.

Single clones were subsequently analyzed for the successful deletion of the *IL-1α* gene using immunostaining, flow cytometry and genomic sequencing of the IL-1α locus (Appendix A). Verified IL-1α KO clones were used in all experiments described below.

### 4.4. Mice

Wild type BALB/c and NOD.SCID (NOD.CB17-*Prkdc^scid^*/NCrHsd) female mice were purchased from Envigo, Israel. NSG (NOD.Cg-*Prkdc^scid^ Il2rg^tm1Wjl^*/SzJ) mice were from The Jackson Laboratory, Israel. IL-1α KO mice and IL-1Ra KO mice (BALB/c background) were generated, as described [86], and kindly provided by Yoichiro Iwakura, Tokyo University. IL-1α KO and IL-1Ra KO mice were housed under specific pathogen-free conditions at the animal facilities of the faculty of Health Sciences, Ben-Gurion University. Animal studies were approved by the Animal Care Committee of Ben-Gurion University. Female, 8-wk-old mice were used in all experiments. 

### 4.5. MTT Proliferation Assay

Proliferation was assessed by using the MTT kit (Merck, Rahway, NJ, USA, 11465007001), via the manufacturer’s instructions. Plates were read (Multiskan Spectrum, ThermoFisher, Waltham, MA, USA) at 570 nm and 630 nm, and OD values were calculated by subtracting 630 nm from 570 nm readings.

### 4.6. Migration Assay

4T1 and 4T1 IL-1α KO cells were seeded in the upper layer of 24 trans-well plates (pore size 8.0 μm, 662638, Greiner bio-one) for 20 h. For identification of migrated cells, membranes were stained with 50 μL of 0.2% crystal violet in 20% methanol for 7 min. After washing, cells were lysed in 300 μL of 10% acetic acid for 10 min, and OD values were measured with a microplate reader at 570 nm.

### 4.7. Attachment Assay

96well plates were coated with 0.32 μL Laminin (0.5 ug/cm^2^ concentration) (BG iMatrix-511, RL511, Biogems, Westlake Village, CA, USA) overnight at 4 °C. Plates were washed in PBS and blocked for 1 h with PBS containing 1% BSA. After the blocking solution was removed, 100 μL of 4T1 cells was added (35,000 cells/well), and plates were incubated for 45 min at 37 °C. The attached cells were fixed and stained with 50 μL of 0.2% crystal violet in 20% methanol for 12 min and subsequently washed with water. Cells were lysed with 50 μL of 10% SDS for 10 min, and OD values were measured at 600 nm in a microplate reader.

### 4.8. Colony Formation Assay

6 well plates were coated with 1.5 mL of 0.5% soft agar (BactoAgar, BD, Franklin Lakes, NJ, USA) in complete DMEM medium and incubated at room temperature for 20 min to solidify the top layer. Subsequently, 1 mL of 0.3% BactoAgar in DMEM medium with 5000 cells was added. The plates were incubated at 37 °C in a humidified atmosphere of 5% CO_2_ incubator. After 3–4 weeks of cell growth in the soft agar, the number of colonies in each well were counted, using a light microscope.

### 4.9. Tumor Growth In Vivo

4T1 cells (2 × 10^5^ cells/50 μL in PBS) were injected orthotopically into a single mammary pad of female mice [52]. The development of local tumors was evaluated using a caliper every 3–4 days, and tumor volume was calculated: (W^2^ × L)/2, where W is width and L is length. At least 5 mice per experimental group were sacrificed at various time points post-injection in most experiments. Subsequently, local tumors, lungs and spleens were removed for further analysis. Lung metastatic nodules were assessed by calculation of metastasis by inverted microscope after tissue fixation with Bouin’s solution (Merck, HT10132, NJ, USA) overnight. Micro metastases were counted in 6 random fields under light microscope after H&E staining.

### 4.10. Immunotherapy

4T1 cells (2 × 10^5^ cells/50 μL in PBS) were injected orthotopically into a single mammary pad of female mice. Twice a week, mice were treated either with neutralizing Abs against IL-1α, 200 mg per mouse (RD001; XBiotech) or an isotypic control. 

### 4.11. Real-Time Quantitative PCR (qPCR)

RNA was extracted from 4T1 cells or from homogenized tissue samples using the ISOLATE II RNA Mini Kit (Bioline, London, England, UK). The quality and total amount of RNA was determined using a ND-1000 spectrophotometer (ThermoFisher Scientific, Waltham, MA, USA). 1 μg of RNA was converted to cDNA using the qScript cDNA synthesis kit (QuantaBio, Beverly, MA, USA), according to the manufacturer’s instructions. Real-time qPCR was performed (Applied Biosystems, Waltham, MA, USA) using SYBR^®^ Green PCR master mix (Applied Biosystems). See Appendix A for the list of primers used. Gene expression was normalized to the *HPRT* gene.

### 4.12. mRNA Sequencing

RNA was extracted from 4T1 cells using the ISOLATE II RNA Mini Kit (Bioline, 52073), and 3’ mRNA sequencing (RNAseq) was performed at The Nancy and Stephen Grand Israel National Center for Personalized Medicine (G-INCPM), Weizmann Institute of Science, Israel. The results were analyzed using NeatSeq-Flow [87].

### 4.13. ELISA

Tumor tissue samples (100 mg) were treated with 0.5% TRITON-X100 in PBS and homogenized. After centrifugation, supernatants were collected and frozen. The following ELISA kits were used according to the manufacturer’s instructions: DY400, DY410, DY1679, DY406 (R&D Systems, Minneapolis, MN, USA), 900-M47, 900-T53 (Peprotech, Rocky Hill, NJ, USA). The cytokine concentration was normalized to the total protein concentration, and measured using a BCA Protein Assay Kit II (K813-2500, BioVision, Milpitas, CA, USA). 

### 4.14. Immunofluorescence Imaging

4T1 cells were cultured in Ibidi Glass Bottom µ-Slides. The cells were fixed with 4% paraformaldehyde, permeabilized with 0.5% Triton X-100, and stained with the appropriate primary and secondary antibodies (Appendix A), as described [20,23]. Hoechst 33,342 (ThermoFisher, 62249, Waltham, MA, USA) was used for nuclear staining. Slides were examined under the Olympus FV-1000 confocal system (×60 magnification).

### 4.15. Immunohistochemistry (IHC) and Hematoxylin-Eosin (H&E) Staining

Tissue samples were embedded in paraffin. Paraffined tissues were cut into 4-micron microtome sections. Samples were stained with HAEMATOXYLIN/Eosin (LE-3801582E, LE-3801601E Leica biosystems) according to a standard protocol. Immunostaining was performed as previously described [24], using indicated first antibodies (Appendix A) and the Universal ImmPRESS kit (Vector Laboratories, Burlingame, CA, USA) as secondary antibodies.

The visualization was performed using the 3 amino-9-ethylcarbazole (AEC) substrate kit (SK-4205, ImmPACT, West Hills, CA, USA) and counterstained with hematoxylin. Sections were examined under a widefield microscope.

### 4.16. Flow Cytometry (FC)

Primary tumors were obtained from mice and digested using a mixture of enzymes [1 g per 100 mL of collagenase type IV, 20,000 units per 100 mL of DNase type IV and 1 mg/mL of hyaluronidase type V (Sigma–Aldrich, St. Louis, MO, USA) at 37 °C for 20–30 min], as described by [52]. Single cell suspensions from tumors were analyzed by FACS or used for RNA isolation. Supernatants from digested tumors were collected and stored at −20 °C for subsequent evaluation.

Spleens and lymph nodes were cut into small pieces and crushed with a syringe plunger, and then passed through a 70 μm filter to remove any remaining large particles from the single cell suspension. Tumor samples were dissociated into single cell suspensions using the Triple Enzyme Mix (Collagenase 1 mg/mL, Hyaluronidase 100 μg/mL, DNase 20 mg/mL), as described in [52]. Cells were blocked with CD16/CD32 monoclonal antibody (eBiosience, #14-0161-82) and stained with specific antibodies (Appendix A). Samples were analyzed using either Gallios or CytoFlex systems (Beckman Coulter, Brea, CA, USA). Datasets were analyzed using FlowJo V10.

### 4.17. Statistical Analysis

Statistical analyses were done using GraphPad Prism V8. Each experiment was performed 3–5 times. For in vivo experiments, 4–10 mice per group per time point were used. A mean value with the standard error of mean (SEM) was calculated for each data point. Significant differences between the results were determined using either unpaired 2-tailed t-test or 2-way ANOVA.

## Figures and Tables

**Figure 1 ijms-25-03950-f001:**
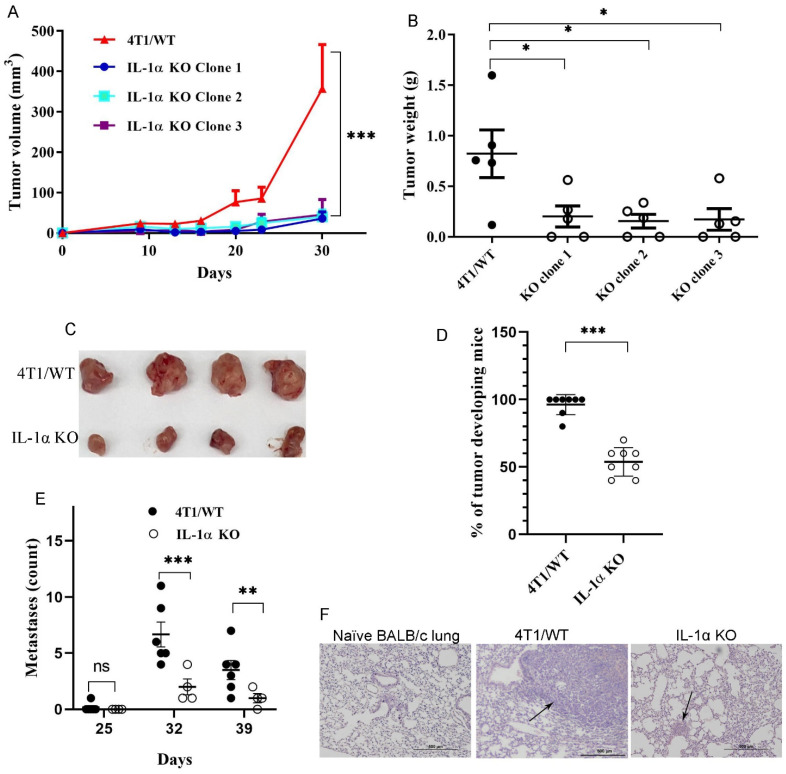
IL-1α knockout in 4T1 cells inhibits tumor development in vivo. Cas9-expressing 4T1 (4T1/WT) or (4T1 IL-1α KO) cells (2 × 10^5^ cells/mouse in 50 μL PBS) were orthotopically injected into the mammary fat pad of BALB/c mice. (**A**) The dynamics of tumor development following injection of either 4T1/WT cells or one of the three distinct 4T1 IL-1α KO clones—a representative experiment with five mice per group per time point. (**B**) Weight of tumors harvested on day 30 post-injection. 4T1 IL-1α KO clone 2 cells were used in experiments shown in (**C**–**F**). (**C**) Representative photographs of tumors harvested on day 30. (**D**) Percentage of tumor-bearing mice following injection of either 4T1/WT or 4T1 IL-1α KO cells on day 30 post-injection; each point represents a separate injection experiment with 5–10 mice per group. (**E**) Metastasis count in lungs of tumor-bearing mice harvested at indicated time points—results of a representative experiment with six mice per group per time point. (**F**) Representative images of H- and E-stained lung tissue sections harvested from either naïve mice or tumor-bearing mice. The arrows point at metastatic foci. The results are shown as average ± SEM. Significant differences are indicated (2-way ANOVA (**A**) or unpaired two-tailed *t*-test (**B**,**D**,**E**)), * *p* < 0.05, ** *p* < 0.01, *** *p* < 0.001).

**Figure 2 ijms-25-03950-f002:**
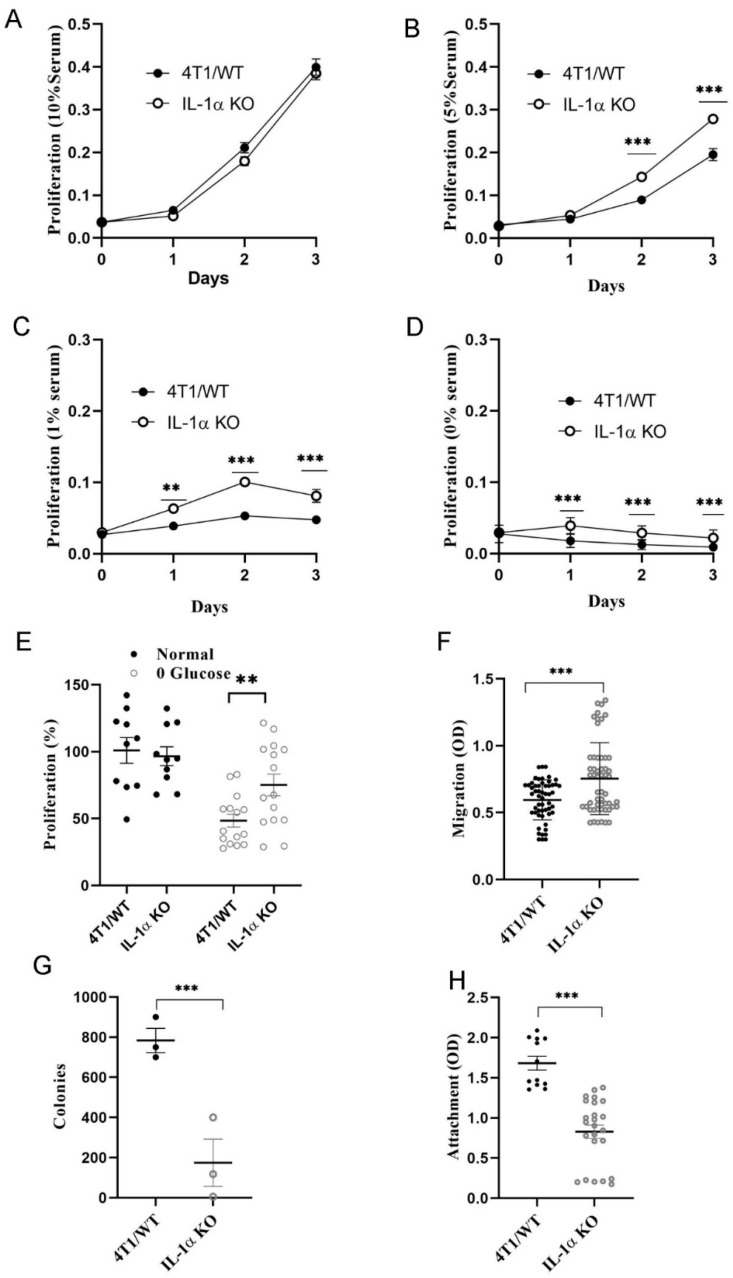
IL-1α expression affects tumorigenicity of 4T1 cells in vitro. (**A**–**D**) 4T1/WT and 4T1 IL-1α KO (clone 2) cells were seeded into 96well plates (2000 cells per well) in 100 μL media and incubated at 37 °C. The proliferation rate was evaluated using the MTT-based assay at different time intervals (1, 2 and 3 days). (**E**) 4T1/WT and 4T1 IL-1α KO cells were grown in either complete medium or in glucose-free medium for 3 days and proliferation rate was assessed, as indicated. Cells (40,000/well) were seeded in transwell plates, as described in Methods, and migration was assessed (**F**); For colony formation assay, cells (5000/well) were incubated in soft agar (see Methods) (**G**); 35,000 cells/well were seeded into laminin-coated plates and cell attachment was assessed (see Methods) (**H**). The results are shown as average ± SEM, from at least three independent experiments. Significant differences are indicated (unpaired two-tailed *t*-test, ** *p* < 0.01, *** *p* < 0.001).

**Figure 3 ijms-25-03950-f003:**
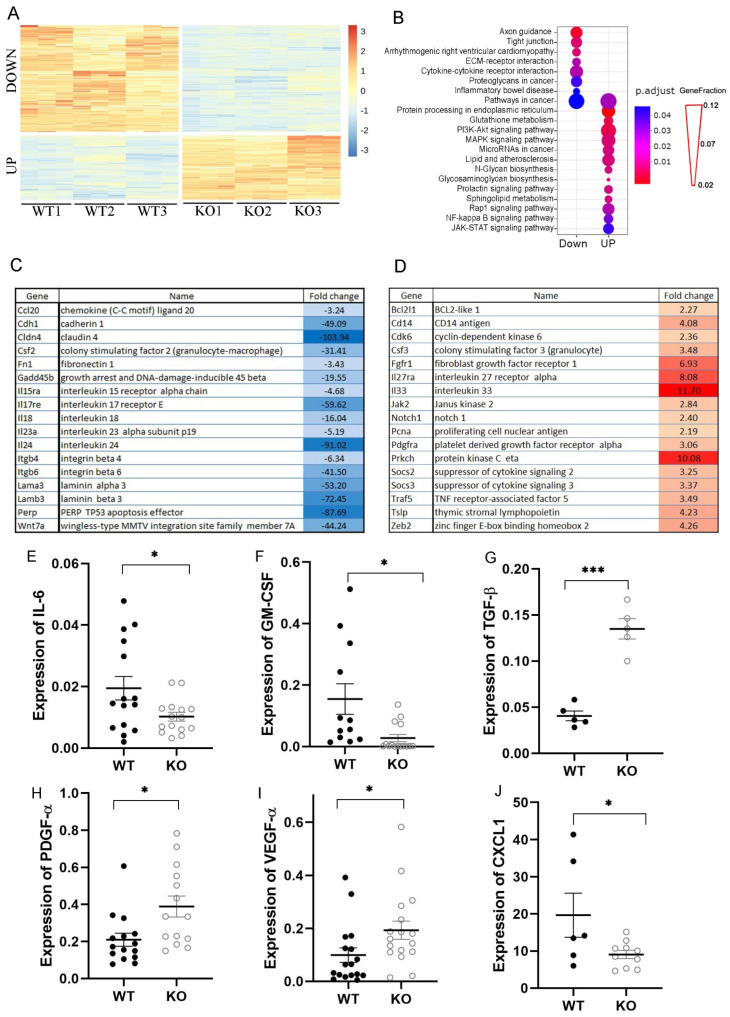
Effect of IL-1α knockout on gene expression in 4T1 cells. (**A**) Heatmap of differentially expressed genes in three Cas9-expressing WT clones (WT1–3) and three 4T1 IL-1α knockout clones (KO1–3). The genes were clustered to either downregulate or upregulate in 4T1 IL-1α KO cells, compared to 4T1/WT cells. (**B**) KEGG pathway enrichment analysis of the significantly upregulated and downregulated genes. The circle size for each pathway corresponds to the fraction of genes out of all affected genes (GeneFraction) that are linked to this pathway. Selected genes linked to the enriched pathways are listed in (**C**) (downregulated) and (**D**) (upregulated). Expression of indicated cytokines and chemokines in either 4T1/WT (WT) or 4T1 IL-1α KO (KO) cells was further assessed by qPCR (**E**–**J**), normalized to HPRT1 as a reference gene. The results are shown as average ± SEM, experiments were performed on at least two separate cultures/lysates. Significant differences are indicated (unpaired two-tailed *t*-test, * *p* < 0.05, *** *p* < 0.001).

**Figure 4 ijms-25-03950-f004:**
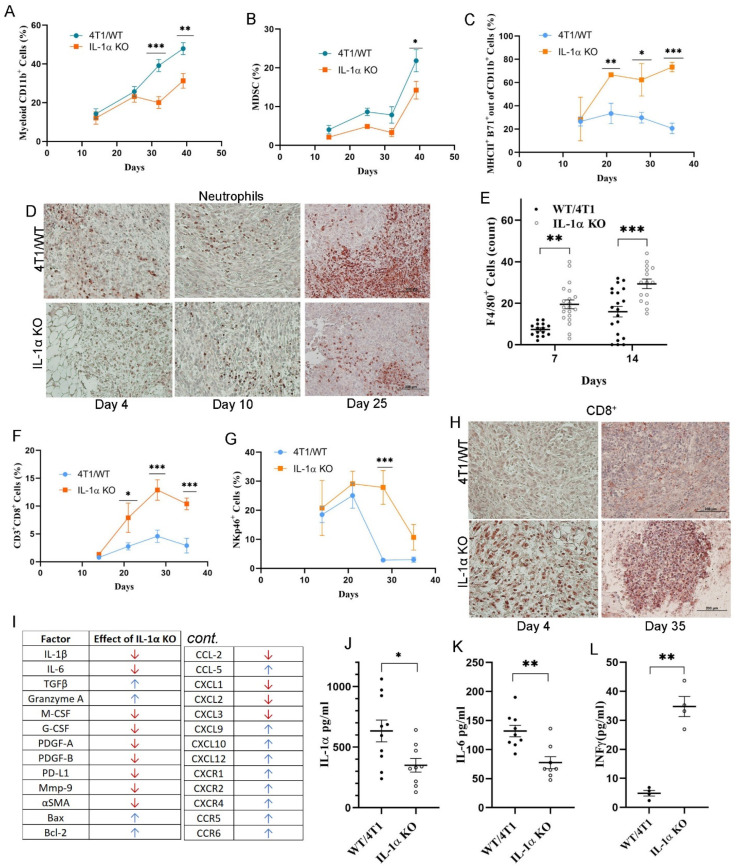
Tumor-derived IL-1α affects immune cellular and molecular composition in the tumor microenvironment. Results of a representative experiment with five (for 4T1/WT) and seven (for IL-1α KO) mice per group per time point are shown in (**A**–**G**). 4T1/WT and 4T1 IL-1α KO cells (clone 2) were injected, as described above. Tumors were harvested at indicated time points post-injection. (**A**) Percentage of CD11b^+^ cells out of all live cells in tumor tissue (FC); (**B**) Percentage of MDSC (defined as a combination of MHC–II^−^CD11b^+^Ly6G^+^Ly6C^low^ and MHC–II^−^CD11b^+^Ly6G^−^Ly6C^high^) out of all live cells in tumor tissue (FC); (**C**) Percentage of activated APCs (CD11b^+^MHCII^+^ B71^+^) among total CD11b^+^ cells in tumor tissue (FC); (**D**) Representative images of tumor tissue sections harvested at indicated time points post-injection and stained for Neutrophil Elastase; (**E**) Amounts of F4/80^+^ cells in tumor tissue on day 35 (IHC); (**F**) Percentage of CD3^+^CD8^+^ cells among all live cells in tumor tissue (FC); (**G**) Percentage of NKp46^+^ cells among live cells in tumor tissue (FC); (**H**) Representative images of tumor tissue sections harvested at indicated time points post-injection and stained for CD8; (**I**) Summary of the genes affected by knockout of IL-1α in 4T1 cells (up or downregulation). Tumor tissues were resected on day 35, and analyzed by qPCR. (**J**–**L**) Amounts of IL-1α (**J**), IL-6 (**K**), IFNγ (**L**) in lysates from tumor tissues resected on day 35 and assessed by ELISA—each point represents an individual tumor. The results are shown as average ± SEM. Significant differences are indicated (unpaired two-tailed *t*-test, * *p* < 0.05, ** *p* < 0.01, *** *p* < 0.001).

**Figure 5 ijms-25-03950-f005:**
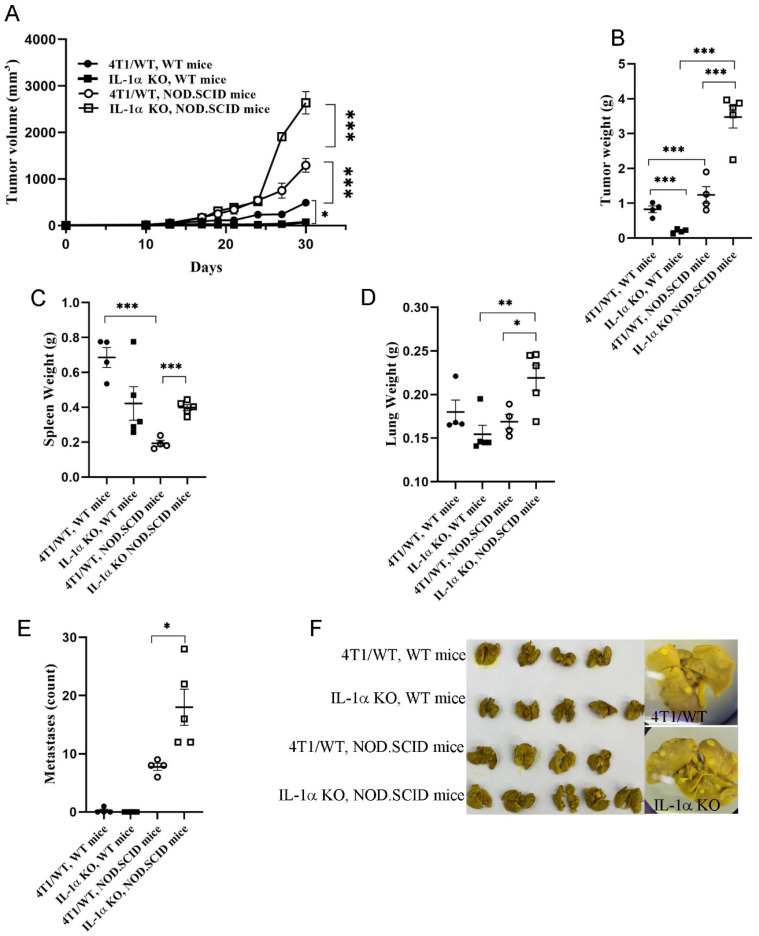
A functional immune system of the host is critical for inhibition of tumor development in mice engrafted with 4T1 IL-1α KO cells. A representative experiment with at least four mice per group per time point is shown. 4T1/WT and 4T1 IL-1α KO cells (clone 2) were injected, as described above, into either BALB/c (WT) or NOD.SCID mice (**A**) The dynamics of tumor development following injection of indicated tumor lines into NOD.SCID or BALB/c mice. Weights of tumors (**B**), spleens (**C**) and lungs (**D**) harvested on day 30 post-injection. (**E**) Metastasis count in lungs of tumor-bearing mice harvested on day 30 post-injection. (**F**) Representative images of lungs harvested on day 30 post-injection. Each experiment included six mice per group, repeated twice. The results are shown as average ± SEM. Significant differences are indicated (two-way ANOVA (**A**) or unpaired two-tailed *t*-test (**B**–**E**), * *p* < 0.05, ** *p* < 0.01, *** *p* < 0.001).

**Figure 6 ijms-25-03950-f006:**
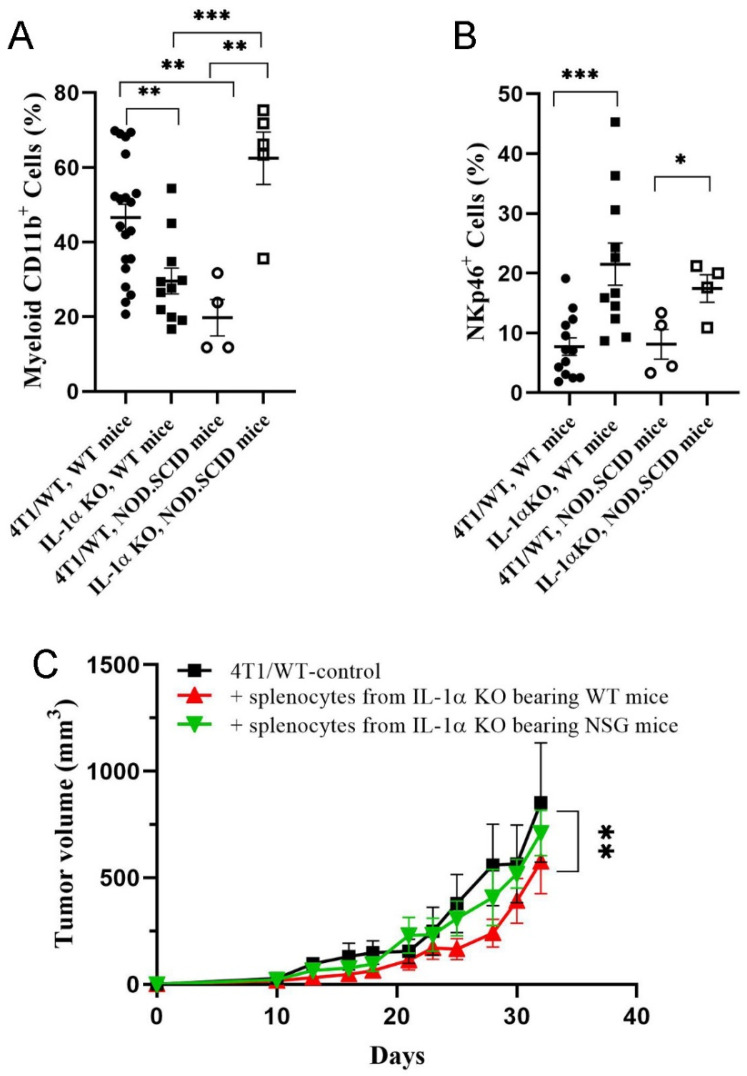
Effects of host immune deficiency on immune cellular composition in the tumor microenvironment. 4T1 or 4T1 IL-1α KO cells (clone 2) were injected, as described above, into either BALB/c (WT) or NOD.SCID mice—at least four mice per group. The tumors and spleens were harvested on day 30 post-injection. (**A**) Percentage of CD11b^+^ cells among total live cells in tumor tissue (FC). (**B**) Percentage of NKp46^+^ cells among total live cells in tumor tissue (FC). (**C**) WT mice were injected with splenocytes from 4T1 IL-1α KO tumor bearing WT or NSG mice and inoculated with 4T1/WT cells. The dynamics of tumor development following inoculation are shown relative to mice untreated with splenocytes (Winn assay). The results are shown as average ± SEM. Significant differences are indicated (unpaired two-tailed *t*-test (**A**,**B**) or two-way ANOVA (C), * *p* < 0.05, ** *p* < 0.01, *** *p* < 0.001).

**Figure 7 ijms-25-03950-f007:**
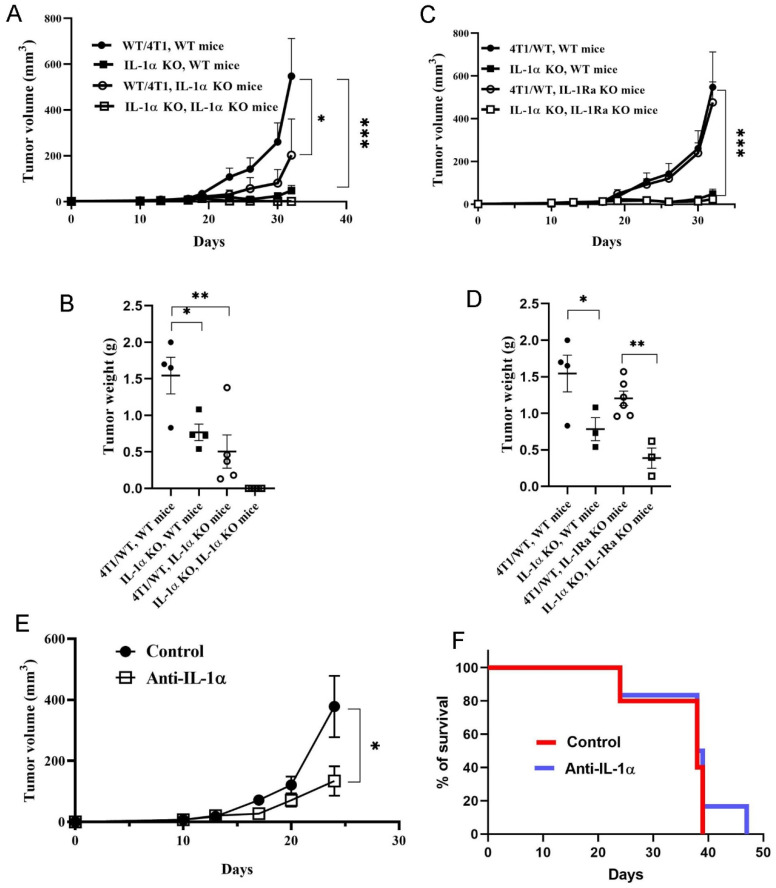
Both the microenvironment-derived and tumor-derived IL-1α affect 4T1-induced tumor development in mice. 4T1/WT and IL-1α KO cells (clone 2) were injected, as described above, into WT mice (**A**–**D**), IL-1α KO mice (**A**,**B**) or IL-1Ra KO mice (**C**,**D**)—at least four mice per group. The dynamics of tumor development are shown in (**A**,**C**); weights of tumors harvested on day 32 post-injection are shown in (**B**,**D**). (**E**) Influence of IL-1α neutralization on 4T1 tumor growth. Tumor- bearing BALB/c mice were treated with either isotype Abs or anti-IL-1α Abs twice a week from day 0 post 4T1 cell injection, and tumor volume was measured every 3–4 days, as described. (**F**) Survival of mice treated with anti-IL-1α Abs. The results are shown as average ± SEM. Significant differences are indicated (two-way ANOVA (**A**,**B**,**E**,**F**) or unpaired two-tailed *t*-test (**C**,**D**), * *p* < 0.05, ** *p* < 0.01, *** *p* < 0.001).

**Figure 8 ijms-25-03950-f008:**
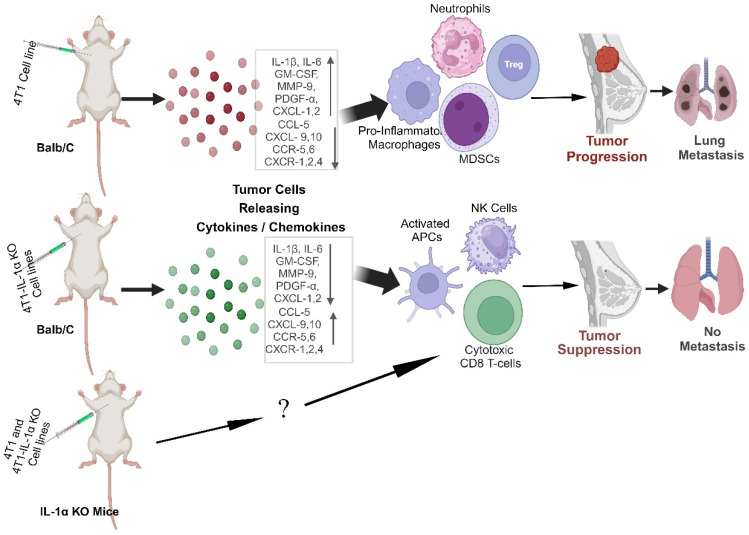
Schematic representation of the mechanism underlying the distinct immune responses against 4T1/WT and 4T1 IL-1α KO cells. IL-1α expression actively influences the immune response by affecting production of factors that attract immune cells to the tumor. The TME of 4T1/WT is enriched in MDSC, neutrophils, and pro-inflammatory macrophages, leading to tumor progression and metastasis. The TME of 4T1 IL-1α KO cells is enriched in cytotoxic cells and activated MHCII^+^ APCs, leading to an effective anti-tumor immune response and tumor suppression. Elimination of host-derived IL-1α also leads to tumor suppression. It remains to be determined if the effects of host-derived IL-1α on the TME parallel the effects of tumor-derived IL-1α.

## Data Availability

The datasets used and/or analyzed during the current study are available from the corresponding author on reasonable request.

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
