# Peer review of "Tumor Cell-Associated IL-1α Affects Breast Cancer Progression and Metastasis in Mice through Manipulation of the Tumor Immune Microenvironment"

_ijms, 2024, doi:10.3390/ijms25073950_

Round 1

Reviewer 1 Report

Comments and Suggestions for Authors

Krishnamohan et al., in this paper, describe the role of IL-1a in the pathogenesis of triple-negative breast cancer. This paper is well written, experiments are well executed with adequate controls, methods are described in adequate detail and the conclusions are well supported by the presented data. I think it is a great manuscript with important findings and should be accepted for publication after addressing the following comment. 

Traditional wisdom suggests that having a more proinflammatory microenvironment tends to better support tumor clearance. IL-1a KO reduced the secretion of inflammatory molecules and increased the secretion of anti-inflammatory molecules (TGF-b). Such stimuli tend to polarize TAM towards M2 (pro-tumor phenotype) and also suppress the function of the effectors such as T and NK cells. Did the authors examine the phenotypes of these cells in the TME?

Author Response

Thank you very much for the time and effort invested in reviewing our manuscript. Please find the response to your comments below.

  1. Traditional wisdom suggests that having a more proinflammatory microenvironment tends to better support tumor clearance.

As it happens, traditional wisdom is occasionally wrong. Inflammation is considered a hallmark of cancer. Chronic inflammation is notoriously pro-carcinogenic. Inflammatory microenvironment positively influences tumor growth, both directly by the factors secreted by the immune cells on the tumor cells, and indirectly through their effect on cells within the TME – stroma, vasculature, etc. We cite several references describing the role of inflammation in cancer progression.

  1. IL-1a KO reduced the secretion of inflammatory molecules and increased the secretion of anti-inflammatory molecules (TGF-b). Such stimuli tend to polarize TAM towards M2 (pro-tumor phenotype) and also suppress the function of the effectors such as T and NK cells. Did the authors examine the phenotypes of these cells in the TME?

Despite an increased level of TGF-b in IL-1α KO tumors, our results show a significant accumulation of MHCII+B7+ F4/80+ cells, which is a marker of M1 macrophages. This could be explained by the increased levels of INFγ (now shown in Fig.4L), which strongly promotes M1 polarization and activation of macrophages.

Reviewer 2 Report

Comments and Suggestions for Authors

In this manuscript, authors have evaluated the effects of IL-1a on breast cancer progression and metastasis. Compared to wild-type (WT) 4T1 tumors, IL-1a knockout (KO) 4T1 tumors showed decreased tumor growth in immunosufficient mice while an increased tumor growth in immunodeficient mice. This suggests that tumor-associated IL-1a negatively affects the host immune response and results in increased tumor growth and metastasis. Authors showed that targeting IL-1a results in reduced tumor progression.

A few suggestions and concerns are provided below:

1.    Results:

a.  Lines 83-85: It is not clear under what conditions ELISA analysis was performed. It will be good to include these data in the supplementary figure with a positive control for IL-1a.

b.    Figure S1B: Refer S1B with S1E and S1F that confirm the KD of IL1-a rather than with S1A that shows its nuclear localization.

c.  Data shown in Figure 2 is contradictory to each other and needs more clarification and explanation. A few examples: while there is an increase in proliferation of IL-1a KO tumor cells, there is reduced colony formation and less tumor growth compared to IL-1a WT cells; Their migration is higher; Is less CF due to their less adherence?; These data show that IL-1a KO 4T1 are more tumorigenic- therefore, authors should explain various possibilities here such as- in vivo there are other cells that express IL-1a (like endothelial cells and immune cells) in IL-1a KO tumor bearing mice or other factors that may be affected by lack of IL-1a signaling in tumor cells may play a role in observed delay in tumor growth.

d.    It is not clear how the cytokines and chemokines shown in Fig. 3E-J and Fig. S2 are selected for real-time PCR (these are not listed in the Figure 3C-D except PCNA and TSLP).

e.    It will be important to compare the results shown in Figure 4I-K from tumors with in vitro results from cells as this may explain (or provide potential reasons) the observed conflicting results such as increased proliferation of IL-1a KO but reduced tumor growth.

f.    Figure 7E: To show the role of host versus tumor IL-1a in tumor development, authors should include groups where host is IL1a KO bearing IL-1a WT tumors or host is WT bearing IL1a KO tumors, neutralize IL-1a, determine the effects on tumor growth and compare with the tumor growth with WT 4T1 cells in WT mice.

g.    Authors should provide some relevance to the data presented in the results throughout the results section. As an example, authors mention on lines 157-158 “we found that knockout of IL-1α in 4T1 cells affected the expression of several cytokines, chemokines, and other factors associated with tumor growth and progression.” What does their up- or down-regulation mean, ex. for tumor growth?

In general, a statement summarizing the most important observations/outcomes at the end of each result section will be helpful to understand the findings.

2.    Discussion:

a.    Lines 399-401: This statement is inaccurate and should be corrected. As one would expect, implantation of either IL-1a WT or KO tumors in WT mice resulted in reduced tumor growth than their implantation in immunodeficient mice. IL-1a KO cells in WT mice grow slower than IL-1a WT cells in WT mice OR IL-1a KO cells in immunodeficient mice. For making the conclusion in the next statement, IL-1a KO cells in WT mice should be compared with IL-1a KO cells in immunodeficient mice.

b.    The paragraph from lines 404 to 417 is not clear. Why TNBC should grow faster in the absence of tumor associated IL-1a in immunodeficient mice. Even if IL-1a is needed for recruitment and activation of APCs, in the immunodeficient mice there are no T cells to have cytotoxic effects or to form memory so effects on tumor growth should still remain the same with or without tumor-associated IL-1a in these mice. Therefore, other possibilities for these observations should be discussed. Further, it is not clear how authors made a conclusion that IL-1a of tumor origin plays a dual role in TNBC development from these data.

c.     Authors should discuss that absence of IL-1a either in the host cells or in the tumor cells results in similar tumor growth while their absence in both results in reduced or no tumor growth. In this regard, it will be important to include the role of host versus tumor IL-1a in tumor development in Fig. 8. Authors may also want to show targeting of IL-1a to reduce tumor progression/metastasis in this figure.

d.    The last statement in the discussion is not supported by the data shown in the manuscript. As mentioned above, targeting IL-1a in both cancer cells and in the host should result in most pronounced effect on tumor growth (not just targeting it in cancer cells).

In addition, minor concerns and suggestions are:

1.    Figure 1 legend: Title- remove “4T1-induced.”; 1D: mention the time point; 1F: what do arrows show; Correct that unpaired t-test was performed for B, D, and E (not F).

2.    Figure 2 legend: In the Figure 2A-D, X-axis is shown in days, so please make the units consistent either as days or hours.

3.   It is important to provide some more experimental details (may be given in the figure legends also so it becomes easier to follow), especially number of times experiments were repeated, data shown is from one individual experiment or average of more than one experiments, time points when analysis was performed, number of replicates used in each experiment, t-test used is one-tailed or two tailed etc.

4.  Supplementary tables 1-3 are referred in the methods, since results section is before methods section, it may be better to re-number the tables (so S4 and S5 referred in the results as tables S1 and S2 respectively).

5.  Figure 3B: Mention/show what the size of circles represents.

6.  Please refer Figures 4C, 4F, 4G, S3E, S3F in the results.

7.  Show individual values of replicates in all the bar graphs.

8.  Line 185: Change to Fig. 4F instead of 4H.

9.  Figure S3F: Why there is a significant decrease in MDSCs at day 32 and then their frequency go up again in spleen?

10.  Figure 4C: Show the legend for two groups as shown in 4A and 4B.

11. Figure 4: Write the number of mice from where tumors and spleens (supplementary figures) were taken and data shown for cells/cytokines etc. in all the figures (line graphs) where it is missing.

12.  Figure 6 legend: Remove ‘D’ after 2-way ANOVA as Fig. 6D is not there.

13.  Figure 7 legend: Provide statistical methods used for 7E and 7F.

14.  Provide version of GraphPad Prism in the methods.

Author Response

Thank you very much for the time and effort invested in reviewing our manuscript. All your comments have been addressed. Please find the response to your comments below.

  1. Lines 83-85: It is not clear under what conditions ELISA analysis was performed. It will be good to include these data in the supplementary figure with a positive control for IL-1α.

The sentence in question has been written poorly. Thank you for noticing our mistake. Only the levels of IL-1α in supernatants were measured by ELISA. The mistake has been corrected (now lines 109-111). The requested figure is included (S1G). Supernatants from cultured peritoneal macrophages were used as a positive control.

2. Figure S1B: Refer S1B with S1E and S1F that confirm the KD of IL1-αrather than with S1A that shows its nuclear localization.

Corrected.

3. Data shown in Figure 2 is contradictory to each other and needs more clarification and explanation. A few examples: while there is an increase in proliferation of IL-1αKO tumor cells, there is reduced colony formation and less tumor growth compared to IL-1α WT cells; Their migration is higher; Is less CF due to their less adherence?; These data show that IL-1 α KO 4T1 are more tumorigenic- therefore, authors should explain various possibilities here such as- in vivo there are other cells that express IL-1 α (like endothelial cells and immune cells) in IL-1 α KO tumor bearing mice or other factors that may be affected by lack of IL-1 α signaling in tumor cells may play a role in observed delay in tumor growth.

We addressed this issue extensively, both in Results (lines 157-163, 192-214) and Discussion (lines 409-427, 460-479, 487-492, 527-531).

4. It is not clear how the cytokines and chemokines shown in Fig. 3E-J and Fig. S2 are selected for real-time PCR (these are not listed in the Figure 3C-D except PCNA and TSLP).

Addressed in lines 189-214.

5. It will be important to compare the results shown in Figure 4I-K from tumors with in vitro results from cells as this may explain (or provide potential reasons) the observed conflicting results such as increased proliferation of IL-1a KO but reduced tumor growth.

This issue is addressed now in multiple points through the manuscript, e.g. in lines 159-163, 328-331, 460-479.

  1.  Figure 7E: To show the role of host versus tumor IL-1 αin tumor development, authors should include groups where host is IL1 α KO bearing IL-1a WT tumors or host is WT bearing IL1 α KO tumors, neutralize IL-1 α, determine the effects on tumor growth and compare with the tumor growth with WT 4T1 cells in WT mice.

The main purpose of this experiment was to separate the effects of intracellular and extracellular IL-1α, since neutralization affects only the latter, while KO affects both. This is explained now in lines 368-374. We did not perform neutralization in IL-1α KO settings, because a) it would defeat the purpose of the experiment; and b) tumor developing in these settings is already strongly inhibited. We hope this explanation is satisfactory.

7. Authors should provide some relevance to the data presented in the results throughout the results section. As an example, authors mention on lines 157-158 “we found that knockout of IL-1α in 4T1 cells affected the expression of several cytokines, chemokines, and other factors associated with tumor growth and progression.” What does their up- or down-regulation mean, ex. for tumor growth?In general, a statement summarizing the most important observations/outcomes at the end of each result section will be helpful to understand the findings.

We have added the summarizing statements to all subsections in Results.

8. Lines 399-401: This statement is inaccurate and should be corrected. As one would expect, implantation of either IL-1α WT or KO tumors in WT mice resulted in reduced tumor growth than their implantation in immunodeficient mice. IL-1 α KO cells in WT mice grow slower than IL-1 α WT cells in WT mice OR IL-1 αKO cells in immunodeficient mice. For making the conclusion in the next statement, IL-1 α KO cells in WT mice should be compared with IL-1 α KO cells in immunodeficient mice.

You are absolutely right; the sentence was written incorrectly. It’s been re-written (lines 466-472).

9. The paragraph from lines 404 to 417 is not clear. Why TNBC should grow faster in the absence of tumor associated IL-1a in immunodeficient mice. Even if IL-1 αis needed for recruitment and activation of APCs, in the immunodeficient mice there are no T cells to have cytotoxic effects or to form memory so effects on tumor growth should still remain the same with or without tumor-associated IL-1 α in these mice. Therefore, other possibilities for these observations should be discussed. Further, it is not clear how authors made a conclusion that IL-1 α of tumor origin plays a dual role in TNBC development from these data.

This issue has been addressed and explained in lines 328-331, 474-479.

10. Authors should discuss that absence of IL-1 αeither in the host cells or in the tumor cells results in similar tumor growth while their absence in both results in reduced or no tumor growth. In this regard, it will be important to include the role of host versus tumor IL-1 α in tumor development in Fig. 8. Authors may also want to show targeting of IL-1a to reduce tumor progression/metastasis in this figure

We have made appropriate changes in Fig. 8. A statement about potential therapeutic targeting of IL-1α expression has been added in Discussion. (Lines 534-540).

11.The last statement in the discussion is not supported by the data shown in the manuscript. As mentioned above, targeting IL- 1αin both cancer cells and in the host should result in most pronounced effect on tumor growth (not just targeting it in cancer cells).

Corrected in lines 84, 534-540.

12.Figure 1 legend: Title- remove “4T1-induced.”; 1D: mention the time point; 1F: what do arrows show; Correct that unpaired t-test was performed for B, D, and E (not F).

Corrected

13.Figure 2 legend: In the Figure 2A-D, X-axis is shown in days, so please make the units consistent either as days or hours.

Corrected

14.  It is important to provide some more experimental details (may be given in the figure legends also so it becomes easier to follow), especially number of times experiments were repeated, data shown is from one individual experiment or average of more than one experiments, time points when analysis was performed, number of replicates used in each experiment, t-test used is one-tailed or two tailed etc.

The information has been included in figure legends.

15. Supplementary tables 1-3 are referred in the methods, since results section is before methods section, it may be better to re-number the tables (so S4 and S5 referred in the results as tables S1 and S2 respectively).

Corrected

16.  Figure 3B: Mention/show what the size of circles represents.

Done.

17.  Please refer Figures 4C, 4F, 4G, S3E, S3F in the results.

Corrected

18.  Show individual values of replicates in all the bar graphs.

All bar graphs have been replaced with individual point scatter plots.

19.  Line 185: Change to Fig. 4F instead of 4H.

Corrected

20.  Figure S3F: Why there is a significant decrease in MDSCs at day 32 and then their frequency go up again in spleen?

We have no reasonable explanation for this fluctuation. However, these are the results, and we report them as we got them.

21.  Figure 4C: Show the legend for two groups as shown in 4A and 4B.

Corrected

22. Figure 4: Write the number of mice from where tumors and spleens (supplementary figures) were taken and data shown for cells/cytokines etc. in all the figures (line graphs) where it is missing.

This information has been included in figure legends.

23.  Figure 6 legend: Remove ‘D’ after 2-way ANOVA as Fig. 6D is not there.

Corrected

24.  Figure 7 legend: Provide statistical methods used for 7E and 7F.

Corrected

25.  Provide version of GraphPad Prism in the methods.

Provided

Reviewer 3 Report

Comments and Suggestions for Authors

This study by Krishnamohan M et al., presneted the data on the role of IL-1α in triple negative breast cancer. Authors utilized 4T1 cancer cells-represents human triple negative breast cancer stage 4, with high metastasis. Authors knocked out IL-1α in 4T1, and assessed tumor growth, lung metastasis. Mice transplanted with IL-1α knocked out 4T1 cells showed decreased tumor burden, metastasis, compared with control 4T1 cells injected mice. The authors observed that Tumor microenvironment is dominated by increased infiltration of CD8+ T cells and NKp46+ cells, whereas the control group showed neutrophils nd MDSCs. Authors also utilized immunodeficient NO.SCID mice, which showed an opposite results; tumor growth, metastasis is increased in mice bearing IL-1α knocked out 4T1 cells. Based on these observations, the authors concluded that IL-1α might involved via immune suppressive as well cytotoxic cells.

The dual role of IL-1α is quite interesting, this way the manuscript presented here may be interesting in the tumor immunology filed. 

The following are my concerns which authors may consider addressing them.

Minor:

Abstract. line 21: please write tumor microenvironemnet (TME). Since the abstract is key to attain readers interest, I suggest to improve it by majorly language editing. Highlight the key findings not in generalized statements.

Introduction. Consider adding background information related to immune cell features of human breast cancer, and also murine 4T1. This information will help readers to easily follow and understand about immune cell infiltration.

Results:  

1) Fig S1-F: which clone does this WB represents?

2) Fig 1: I am confused, authors mentioned in line 92 that three clones showed negligible IL-1α expression, but which clone was further used thought the experiments (c, d, e, f ad so on)? Please change data to scatterplots for b, d, e. What is % tumor development? F-represents H&E staining from three groups for a comparison, however, the scale bar is different. Please present images acquired at the same magnification. Scalar must be readable.

3) line 135 are these cells from the WT Balbc mice transplanted with 4T1 cells vs WT+IL-1α KO 4T1 cells. If not, Isn't it simple to mention 4T1 vs IL-1α KO 4T1 

4) Fig 4; Authors need to provide concise but required information to easily understand the figure. Please indicate in the legend that the data in A, B, C by Flow cytometry. If the cell population of B and C calculated as % of total myeloid cells, how did A was represented? % of CD45? please improve this information in the manuscript, ie figure legends, and material and methods. For D and H, please indicate Neutrophils and CD8+ on the top of the panel. Change graph J to scatterplot.

Major:

1) As I understand, the increased number of CD8+ T cells and NKp46+ cells were considered as anti-tumor drivers in mice transplanted with IL-1α KO 4T1. However, authors did not sequentially phenotypes immune cells in the tumors. Eg. authors did not show whether these CD8 cells show increased intracellular expression for IFN-y, Granzyme B, IL-10. TNF-a etc. Without these analysis, authors may not group them as anti-tumor, therefore, the title of the manuscript is not justified. 

2) As stated above, there are some limitations of this study. Therefore, I suggest authors changing the title and modify abstract-particularly line 24-26. And modify the discussion accordingly. 

3) Immune phenotypic of tumor and peripheral blood would have given more insights of TME and recruitment phenomenon. Importantly, Tumor cells RNAseq approach might have been more valuable in this set. 

Overall, though the technical limitations and small study design gaps, this manuscript sufficiently provided convincing and interesting data. This manuscript can be accepted after minor revisions.  

Author Response

Thank you very much for the time and effort invested in reviewing our manuscript. All your comments have been addressed. Please find the response to your comments below.

  1. line 21: please write tumor microenvironemnet (TME). Since the abstract is key to attain readers interest, I suggest to improve it by majorly language editing. Highlight the key findings not in generalized statements.

Abstract has been modified accordingly.

  1. Introduction. Consider adding background information related to immune cell features of human breast cancer, and also murine 4T1. This information will help readers to easily follow and understand about immune cell infiltration.

We have added the relevant information with references to Introduction (Lines 71-83). References to the background on 4T1 cells have been cited (Lines 104-106).

  1. Fig S1-F: which clone does this WB represents? Fig 1: I am confused, authors mentioned in line 92 that three clones showed negligible IL-1α expression, but which clone was further used thought the experiments (c, d, e, f ad so on)?

The information about the clones used is added to figure legends.

  1. Please change data to scatterplots for b, d, e.

All bar graphs have been replaced with individual point scatter plots.

  1. What is % tumor development?

Corrected to “% of tumor developing mice”. The subject is addressed in lines 126-127.

  1. F-represents H&E stainingfrom three groups for a comparison, however, the scale bar is different. Please present images acquired at the same magnification. Scalar must be readable.

Corrected to the same scale.

  1. line 135 are these cells from the WT Balbc mice transplanted with 4T1 cells vs WT+IL-1α KO 4T1 cells. If not, Isn't it simple to mention 4T1 vs IL-1α KO 4T1 

This figure describes in-vitro experiments with cells grown in culture. Wild type 4T1 cells (4T1/WT) are compared with 4T1 IL-1α KO. This notation is used throughout the manuscript.

  1. Fig 4; Authors need to provide concise but required information to easily understand the figure. Please indicate in the legend that the data in A, B, C by Flow cytometry.

Indicated as “(FC)”

  1. If the cell population of B and C calculated as % of total myeloid cells, how did A was represented? % of CD45? please improve this information in the manuscript, ie figure legends, and material and methods.

The information has been added to figure legends.

  1. For D and H, please indicate Neutrophils and CD8+ on the top of the panel. Change graph J to scatterplot.

Corrected.

  1. As I understand, the increased number of CD8+ T cells and NKp46+ cells were considered as anti-tumor drivers in mice transplanted with IL-1α KO 4T1. However, authors did not sequentially phenotypes immune cells in the tumors. Eg. authors did not show whether these CD8 cells show increased intracellular expression for IFN-y, Granzyme B, IL-10. TNF-a etc. Without these analysis, authors may not group them as anti-tumor, therefore, the title of the manuscript is not justified.

These cells were suggested as the drivers behind the observed suppression of tumor growth since malignant cells are usually eliminated by CTLs and NK cells. Actually, we did detect elevation in Granzyme B (Fig.4I) and IFNγ (now shown in Fig.4L) in the tumor tissue. This finding correlates with the increase in percentages of CTLs and NK cell in the tumor.
Regardless, we have modified the title as requested.

  1. As stated above, there are some limitations of this study. Therefore, I suggest authors changing the title and modify abstract-particularly line 24-26. And modify the discussion accordingly.

We have modified the title and Abstract. Additionally, the results were discussed more extensively, the suggested explanations and hypotheses have been re-phrased and elaborated in Discussion.

  1. Immune phenotypic of tumor and peripheral blood would have given more insights of TME and recruitment phenomenon. Importantly, Tumor cells RNAseq approach might have been more valuable in this set. 

We have used RNAseq approach to analyze tumor cells grown in culture only. However, we used flow cytometry to characterize immune cell composition of the TME, while ELISA and qPCR techniques were used to analyze the molecular composition of tumor tissue. The results are shown in Fig.4.

Round 2

Reviewer 2 Report

Comments and Suggestions for Authors

Thanks for answering most of my comments. A few minor concerns:

1.     In response to my earlier comment, “Figure 3B: Mention/show what the size of circles represents”, authors have stated in the figure legend that the circle size represents number of genes. However, authors need to show a legend (similar to the p.adjusted legend) in the figure where each circle size corresponds to a range of number of genes (from the smallest circle size gradually increasing to the largest circle size).

2.     In response to my earlier comment, “Figure S3F: Why there is a significant decrease in MDSCs at day 32 and then their frequency go up again in spleen?”, authors have responded, “We have no reasonable explanation for this fluctuation. However, these are the results, and we report them as we got them.” 

I understand that this is the data authors obtained but do the authors observe the same fluctuation and exactly at the same timepoints in all the repeat experiments (provided at least two independent experiments were performed and data shown is representative of those)? If yes, then this is not something just a fluctuation; if no, then either data from the repeat experiment can be shown or the data from all the repeat experiments can be averaged and shown.

In this regard, as I suggested earlier, it will be good to provide this information, that is how many time experiments were repeated either in each figure legend or in the statistical section of the methods.

3.     Correct the typo for “separate” (on line 399 of the tracked pdf version), remove “injected” (on line 402 of the tracked pdf version), and change to “inoculated BALB/c mice” from “inoculated mice BALB/c” (on line 403 of the tracked pdf version).

Author Response

Thank you again for the time and effort invested in reviewing our manuscript. Your additional comments have been addressed. Please find the responses to your comments below.

  1. In response to my earlier comment, “Figure 3B: Mention/show what the size of circles represents”, authors have stated in the figure legend that the circle size represents number of genes. However, authors need to show a legend (similar to the p.adjusted legend) in the figure where each circle size corresponds to a range of number of genes (from the smallest circle size gradually increasing to the largest circle size).

We have defined precisely the mathematical value represented by the circle size in the figure legend and added a graphical legend in the figure itself as requested.

  1. In response to my earlier comment, “Figure S3F: Why there is a significant decrease in MDSCs at day 32 and then their frequency go up again in spleen?”, authors have responded, “We have no reasonable explanation for this fluctuation. However, these are the results, and we report them as we got them.” I understand that this is the data authors obtained but do the authors observe the same fluctuation and exactly at the same timepoints in all the repeat experiments (provided at least two independent experiments were performed and data shown is representative of those)? If yes, then this is not something just a fluctuation; if no, then either data from the repeat experiment can be shown or the data from all the repeat experiments can be averaged and shown. In this regard, as I suggested earlier, it will be good to provide this information, that is how many time experiments were repeated either in each figure legend or in the statistical section of the methods.

We have modified Figure S3F. It now shows results averaged over 2 experiments instead of a representative from a single one. This time-course experiment was actually performed 3 times, but the time points of the third experiment were slightly different from the other two, so the results could not be averaged over all 3 of them.

Generally, we state in Methods that all experiments have been performed at least 3 (and up to 5) times, and mention in figure legends whether the graphs show a representative experiment or an average over more than one experiment.

  1. Correct the typo for “separate” (on line 399 of the tracked pdf version), remove “injected” (on line 402 of the tracked pdf version), and change to “inoculated BALB/c mice” from “inoculated mice BALB/c” (on line 403 of the tracked pdf version).

The typos have been corrected.